# Thermal crumpling of perforated two-dimensional sheets

David Yllanes [1,2,3], Sourav S. Bhabesh [1,2], David R. Nelson[4] & Mark J. Bowick [1,2]

Thermalized elastic membranes without distant self-avoidance are believed to undergo a crumpling transition when the microscopic bending stiffness is comparable to $kT$, the scale of thermal fluctuations. Most potential physical realizations of such membranes have a bending stiffness well in excess of experimentally achievable temperatures and are therefore unlikely ever to access the crumpling regime. We propose a mechanism to tune the onset of the crumpling transition by altering the geometry and topology of the sheet itself. We carry out extensive molecular dynamics simulations of perforated sheets with a dense periodic array of holes and observe that the critical temperature is controlled by the total fraction of removed area, independent of the precise arrangement and size of the individual holes. The critical exponents for the perforated membrane are compatible with those of the standard crumpling transition.

[1] Department of Physics and Soft Matter Program, Syracuse University, Syracuse, NY 13244, USA. [2] Kavli Institute for Theoretical Physics, University of California, Santa Barbara, CA 93106, USA. [3] Instituto de Biocomputación y Física de Sistemas Complejos (BIFI), 50009 Zaragoza, Spain. [4] Department of Physics and School of Engineering and Applied Sciences, Harvard University, Cambridge, MA 02138, USA. Correspondence and requests for materials should be addressed to D.Y. (email: dyllanes@syr.edu)

Two-dimensional materials such as graphene[1] or MoS$_2$[2] currently enable the experimental study[3] of the mechanical properties of thermalized elastic sheets and a testing ground for many longstanding theoretical and numerical predictions[4–21]. Particularly striking is the possibility of engineering elastic parameters such as the bending rigidity and Young's modulus over broad ranges simply by varying the overall size or temperature of atomically thin cantilevers and springs (see refs. [22,23] for general references on elastic membranes).

Recent work by Blees et al.[24], in addition to demonstrating a 4000-fold enhancement of the bending rigidity relative to its $T = 0$ value, has shown the potential of graphene as the raw ingredient of microscopic mechanical metamaterials. Employing the principles of kirigami (the art of cutting paper), one can construct robust microstructures, thus providing an alternative route to tune mechanical properties, leaving graphene's remarkable electrical properties essentially intact. These results serve as inspiration for further theoretical work on the interplay between geometry and mechanics[25].

A common working model of elastic membranes is the crystalline or polymerized membrane[10], which serves as a useful tool to describe systems ranging from graphene to the spectrin cytoskeleton of red blood cells[26], polymersomes[27] and assemblies of spider silk proteins[28]. In this context, a key theoretical prediction is the existence of a crumpling transition. For low temperatures, the thermal sheet is in a flat phase roughened by flexural phonons, with long-range order in the orientations of surface normals, analogous to the ferromagnetic phase of a spin system. At sufficiently high temperatures, however, thermal fluctuations can disorder the membrane and drive it to a crumpled phase, with only short-range order in the normals. This prediction has been confirmed analytically and numerically, at least for phantom membranes, without distant self-avoidance[9,10,23,29]. Unfortunately, in many materials, the crumpling transition is very far from the experimentally accessible regime: for instance, graphene has a bending rigidity of $\kappa_0 \approx 48\ kT_R$, where $T_R$ is room temperature[11]. For the intact lattice, this corresponds to a crumpling temperature of order $10^4$–$10^5$ K, well beyond the melting point for graphene. Clearly, we need a mechanism to lower the crumpling transition temperature.

We show here that the crumpling temperature can be significantly lowered by altering the geometry and topology of the membrane. In particular, we perform extensive molecular dynamics simulations of crystalline membranes with dense periodic arrays of holes and determine the dependence of the onset of crumpling on the degree of perforation. This dependence is very strong, but can be shown to be a function of a simple control parameter, namely the total fraction of removed area, and independent of the detailed arrangement and size of the individual holes.

## Results

**Our model.** We consider square sheets of size $L \times L$, which for the purposes of computer simulation, we discretize with a tiling of equilateral triangles of side $a = 1$. We use a standard coarse-grained model[30] to compute the elastic energy in the sheet, which is composed of a stretching and a bending term

$$\mathcal{H} = \mathcal{H}_{\text{stretch}} + \mathcal{H}_{\text{bend}}. \tag{1}$$

Stretching is modeled by considering each triangle side as a spring of elastic constant $\epsilon$ and rest length $a$:

$$\mathcal{H}_{\text{stretch}} = \frac{1}{2}\epsilon \sum_{\langle i,j \rangle} \left( r_{ij} - a \right)^2, \tag{2}$$

where the sum is over all pairs of vertices joined by a triangle edge. The bending energy is represented by a standard dihedral interaction between normals,

$$\mathcal{H}_{\text{bend}} = \tilde{\kappa} \sum_{\langle \alpha, \beta \rangle} \left( 1 - \hat{\mathbf{n}}_\alpha \cdot \hat{\mathbf{n}}_\beta \right). \tag{3}$$

Here the sum is over all the pairs of triangles that share a side and $\hat{\mathbf{n}}_\alpha$ is the unit normal to triangle $\alpha$. Note that placing a carbon atom at the center of each triangle provides an approximate atomic model for the elastic modes of graphene on a dual lattice, as long as we choose the bending rigidity and Young's modulus correctly.

The elastic parameters $\epsilon$ and $\tilde{\kappa}$ are directly related to the continuum Young's modulus $\left( Y_0 = 2\epsilon/\sqrt{3} \right)$ and bare bending rigidity $\left( \kappa_0 = \sqrt{3}\tilde{\kappa}/2 \right)$. Normally, when performing a numerical

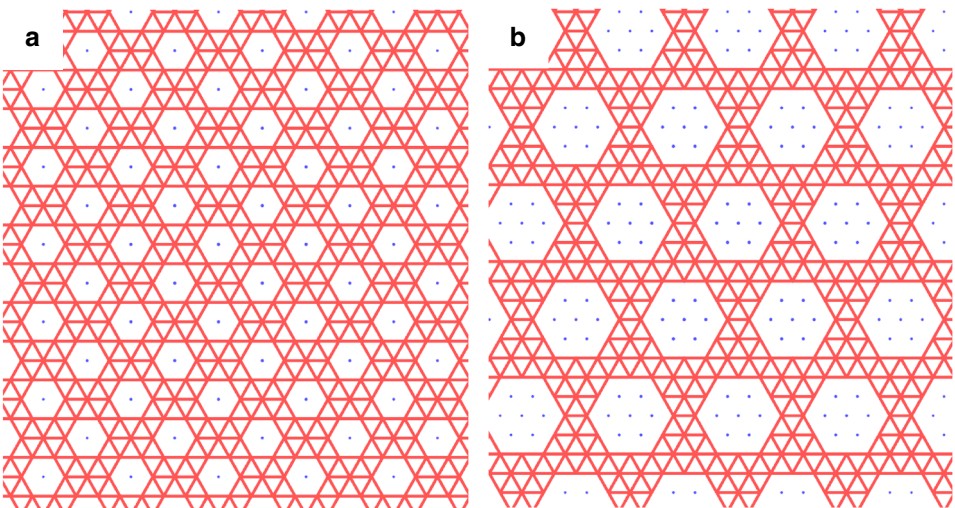

**Fig. 1** Two different arrays of perforations. For clarity, the images show only a small section of the full membrane. We begin by considering a full triangulated sheet and then remove all the nodes in a radius $R$ around its center. This central hole is then repeated periodically throughout the membrane. In the figure, the removed nodes are represented by blue dots (**a**: $R = 1$, **b**: $R = 2$). In the rest of the paper, we will consider patterns of perforations with $R = 1, 2$ and varying spacing between holes (see Supplementary Fig. 1 for a full description of all perforation patterns)

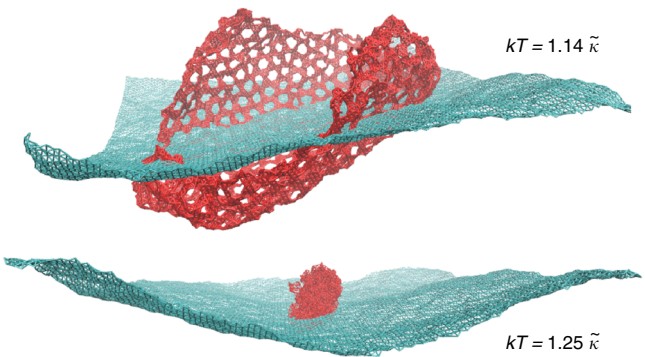

$kT = 1.14\ \tilde{\kappa}$

$kT = 1.25\ \tilde{\kappa}$

**Fig. 2** Snapshots of thermalized configurations. We superimpose the configurations of a pristine sheet (blue) and a perforated sheet with holes of size $R = 2$, in the pattern of Fig. 1b (red) for two values of the temperature (in units of the bending rigidity $\tilde{\kappa}$). In both cases, the full sheet is well into the flat phase, the thermal fluctuations causing just some wrinkling and oscillation. This 10% increase in the temperature, however, triggers a crumpling of the perforated sheet. Both systems have size $L = 100a$. See Supplementary Movies 1 and 2 for animations of these simulations

study (see, e.g., refs. [9,31]) one chooses natural units where $\epsilon = 1$ and $\tilde{\kappa}$ is varied. To better approximate the behavior of materials such as graphene or $MoS_2$, however, we will instead fix the ratio $\epsilon/\tilde{\kappa}$ and vary temperature by changing the $\tilde{\kappa}/kT$ ratio. For graphene, at room temperature, $\tilde{\kappa}/kT \approx 48$[11]. Following refs. [25,32], we will use $\epsilon/\tilde{\kappa} = 1440/a^2$. This corresponds to a Young's modulus about an order of magnitude lower than for real graphene[33,34], in order to facilitate equilibration in our computer simulations. We note, however, that this choice should have only a minor effect on the onset of crumpling, since the degree of order in the normals only depends on $\epsilon/\tilde{\kappa}$ logarithmically[21]. For a sheet of length $L/a = \mathcal{O}(10^2)$, corresponding to a patch of freely suspended graphene, roughly 30 nm on a side, these parameters result in a Föppl von Kármán number of $\nu K = Y_0 L^2/\kappa_0 \sim \mathcal{O}(10^7)$, similar to that of a standard A4 sheet of paper.

**Dense arrays of holes**. We are interested in exploring the effect of a perforated geometry on the rigidity of elastic membranes. To this end we shall compare the physics of the "full" or unperforated sheet described above with that of a sheet with a dense array of holes. We begin by removing the node $i = 0$ situated in the center of the sheet and all the nodes $j$ such that $r_{0j} = |\mathbf{r}_j| < R$. We then repeat this operation periodically throughout the lattice to create a dense lattice of perforations (Fig. 1 and Supplementary Fig. 1). In this paper, we consider arrays of holes of size $R = 1, 2$ with varying spacing. We kept the radius of the hole small relative to the length of the sheet to minimize finite-size effects.

As a first demonstration of the dramatic effect of these perforations, consider the pattern of holes depicted in Fig. 1 —right. Figure 2 compares the equilibrium configurations of this perforated sheet and those of the full membrane for two values of temperature that differ by only 10%. In both cases, the full membrane is deep in the flat phase and exhibits smooth, approximately flat configurations. The perforated sheet, on the other hand, experiences a crumpling transition.

To characterize this transition, it will be useful to consider the radius of gyration of the sheet

$$\mathcal{R}_g^2 = \frac{1}{3N}\sum_{i=1}^{N}\langle\mathbf{R}_i \cdot \mathbf{R}_i\rangle, \quad \mathbf{R}_i = \mathbf{r}_i - \mathbf{r}_{CM}, \qquad (4)$$

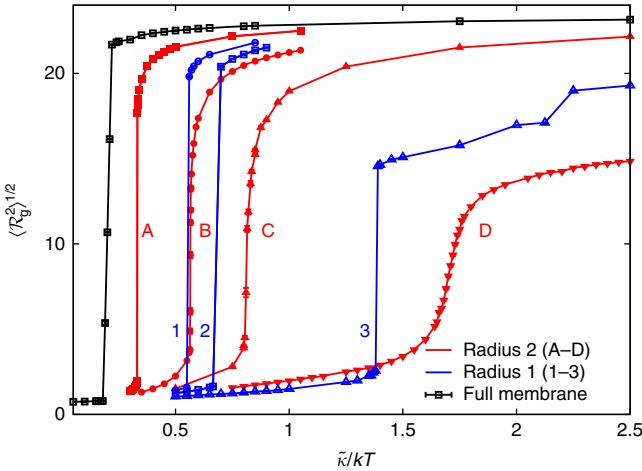

**Fig. 3** Radius of gyration as a function of $\tilde{\kappa}/kT$ for several patterns of perforation. The black curve is the baseline unperforated membrane, which crumples at the highest temperature, $\tilde{\kappa}/kT_c \approx 0.18$. The red curves **A**–**D** are for triangular arrays of perforations of radius $R = 2$ with decreasing spacing between individual holes. The blue curves (1–3) are arrays of perforations with radius $R = 1$. As the spacing between holes is reduced, the crumpling temperature decreases. The full description of the perforation patterns **A**–**D** and 1–3 is given in Supplementary Fig. 1. Data for systems of size $L = 100a$

where $\mathbf{r}_{CM}$ is the position of the center of mass and $\langle\mathcal{O}\rangle$ represents a thermal average. In the flat phase, $\mathcal{R}_g^2 \sim L^{4/d_H}$, with Hausdorff dimension $d_H = 2$, while in the crumpled phase $\mathcal{R}_g^2 \sim \log(L/a)$ ($d_H = \infty$). In the critical region, the Hausdorff dimension has been computed with analytical methods ($d_H = 2.73$[7]) and with numerical simulations ($d_H = 2.70(2)$[31]).

We have plotted $\mathcal{R}_g$ as a function of $\tilde{\kappa}$ for all our perforation patterns in Fig. 3. In blue (red), we represent systems with arrays of holes of radius $R = 1$ ($R = 2$) and a decreasing separation between holes. The black curve provides the baseline value of $\mathcal{R}_g^2$ for the full membrane. We are interested in computing the critical $kT_c/\tilde{\kappa}$ for crumpling in each of these geometries. This can be done by searching for the maximum in the specific heat of the system, which can be computed as[35]

$$C_V = \frac{1}{N}\left(\langle\mathcal{H}^2\rangle - \langle\mathcal{H}\rangle^2\right). \qquad (5)$$

Alternatively, we can consider the $\tilde{\kappa}$-derivative of $\mathcal{R}_g^2$, which can be evaluated as:

$$\frac{d\mathcal{R}_g^2}{d\tilde{\kappa}} = \frac{kT}{\tilde{\kappa}}\left(\langle\mathcal{H}\rangle\left\langle\mathcal{R}_g^2\right\rangle - \left\langle\mathcal{H}\mathcal{R}_g^2\right\rangle\right). \qquad (6)$$

We show these two quantities for two different perforation patterns in Fig. 4. In the thermodynamic limit, the positions of the peaks in $C_V$ and $d\mathcal{R}_g^2/d\tilde{\kappa}$ tend to the same $kT_c/\tilde{\kappa}$ value. For our finite systems, we use the difference in these peak positions for our most perforated membrane (the case where the peaks are most separated) as an estimate of our systematic error in $\tilde{\kappa}/kT_c$.

In principle, one could think that this $T_c$ would depend in a complicated way on the particular spatial arrangement of the holes or on their individual sizes. Fortunately, the reality is much simpler. Indeed, in Fig. 5, we have plotted the $kT_c/\tilde{\kappa}$ for each of the curves in Fig. 3 as a function of the fraction of removed area in the sheet. Given our discretization, this areal fraction is most easily estimated by counting the fraction of remaining dihedrals connecting adjacent triangles, after the holes have been made.

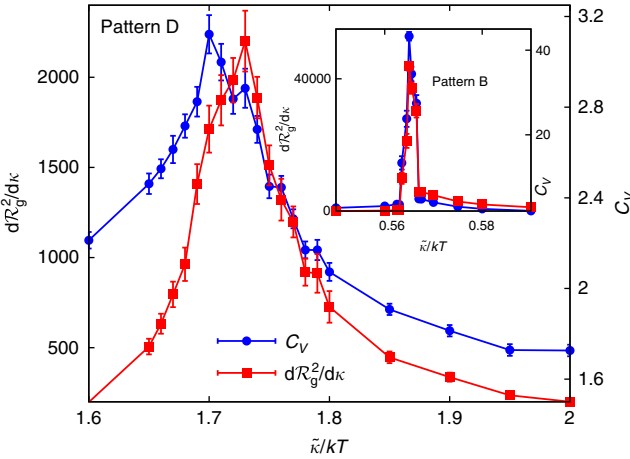

**Fig. 4** Location of the crumpling temperature. We plot the specific heat $C_V$ (right axis) and $\tilde{\kappa}$-derivative of the radius of gyration (left axis) as a function of $\tilde{\kappa}/kT$ for our most perforated system (corresponding to curve D in Fig. 3). The inset shows the analogous plot for a less perforated sheet (corresponding to curve B in Fig. 3), with a much sharper transition (note the different vertical scales of the axes). All error bars represent the standard error of the mean

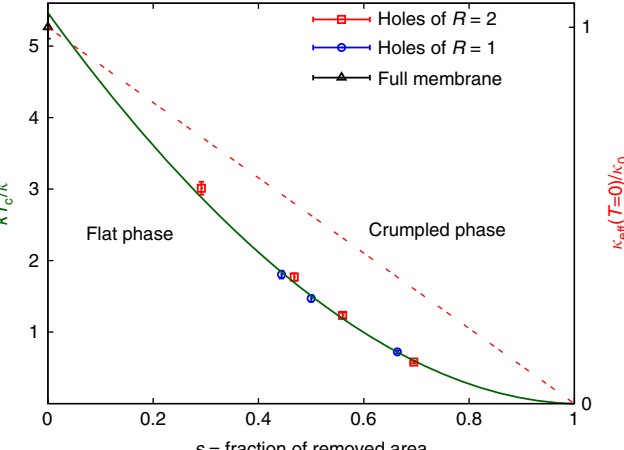

**Fig. 5** Crumpling temperature $kT_c$ as a function of the fraction $s$ of removed area. When plotted against this parameter, the values of $kT_c$ for all eight curves in Fig. 3 collapse to a single smooth function, independent of the size of the individual holes or their precise geometrical arrangement. The curve is a fit to $f(s) = A(1-s)^c$, with $c = 1.93(4)$ and a goodness-of-fit estimator $\chi^2/\text{d.o.f.} = 8.72/6$. On the right-hand vertical axis, we also plot the zero-temperature effective bending rigidity in units of $\kappa_0$ (red dotted line), which is simply linear in $(1-s)$. The error bars represent an estimate of our systematic error, as explained in the text. d.o.f. degrees of freedom

As a function of this dimensionless area fraction, all our $T_c$, including the one for the full membrane, fall on a single smooth curve.

In fact, if we denote by $s$ the fraction of removed area in the perforated sheet, we have found that the following ansatz reproduces our results very accurately:

$$kT_c/\tilde{\kappa} = A(1-s)^c. \tag{7}$$

With our choice of parameters, we obtain a good fit with $c = 1.93(4)$ and $A \approx 5.5$. Notice, in particular, that for our most perforated sheet (where about 70% of the area has been removed), the value of $kT_c$ is reduced by a factor of 10 compared to the full membrane. Extrapolating using Eq. (7), we find that removing 85% of the area in a graphene sheet would bring the crumpling temperature down to about 1600 K. Thus creating "lacey graphene" via, say, laser-ablated holes that remove 85% of the carbon atoms could allow the crumpled regime to be accessed experimentally. We note that the mechanical and electrical properties of free-standing graphene springs with roughly 40% of the material removed were studied in ref. [24].

It is important to note that the observed $kT_c(s)$, Eq. (7), cannot be explained by the effective elastic constants of the perforated sheets $\kappa_{\text{eff}}(T=0)$ and $Y_{\text{eff}}(T=0)$. Indeed, as we explain in Supplementary Note 1, the $T=0$ bending modulus of the perforated sheet, $\kappa_{\text{eff}}(T=0)$, linearly decays with $(1-s)$. Therefore, if the onset of crumpling were simply determined by $\kappa_{\text{eff}}(T=0)$, one would expect $T_c$ to be a linear function of $(1-s)$. Instead, as we obtained in Eq. (7), $T_c \sim (1-s)^{1.93}$, a result which indicates that nontrivial thermal fluctuation effects are responsible. The effective Young's modulus $Y_{\text{eff}}(T=0)$, on the other hand, has a complicated dependence on the details of the perforation pattern (Supplementary Note 1). However, $Y$ only affects the crumpling temperature as a logarithmic correction, see Eq. (9) below. Explaining the observed value of $c = 1.93(4)$ remains, therefore, a theoretical challenge.

**Finite-size scaling**. We have seen that cutting holes in a membrane can induce crumpling at much lower temperatures. We have yet to show, however, that this phenomenon quantitatively

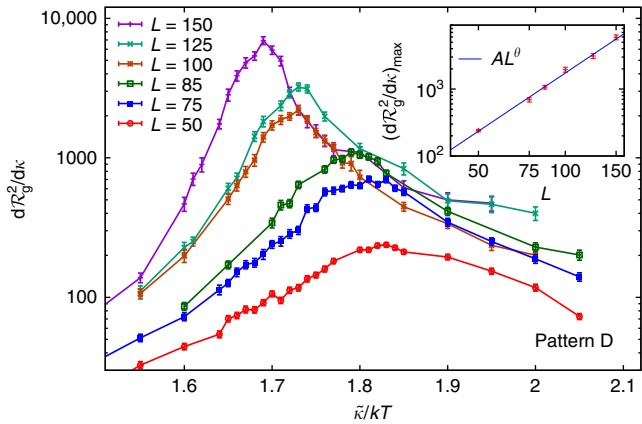

**Fig. 6** Peak of $d\mathcal{R}_g^2/d\tilde{\kappa}$ for our most perforated sheet and six system sizes $L$. Inset: scaling of the height of the peak with an exponent $\theta = 4/d_H + 1/\nu = 2.88(7)$, from a fit with $\chi^2/\text{d.o.f.} = 4.67/4$. The expected value for the crumpling transition[7] is $\theta \approx 2.82$. All error bars represent the standard error of the mean

corresponds to the standard crumpling transition that has been extensively studied for full sheets[7,9,16,17,29,31,35–41]. This can be accomplished by performing a finite-size scaling (FSS) study[42] and finding the universality class of the phase transition. This computation poses two difficulties: on the one hand our simulations cover a very wide range of temperature, rather than concentrating all the numerical effort to increase the precision at the critical region. On the other hand, the presence of the holes creates novel finite-size effects. We begin by considering the FSS of the height of the peak in $d\mathcal{R}_g^2/d\tilde{\kappa}$, which diverges as[31]

$$\left.\frac{d\mathcal{R}_g^2}{d\tilde{\kappa}}\right|_{\max} \sim L^{4/d_H + 1/\nu}, \tag{8}$$

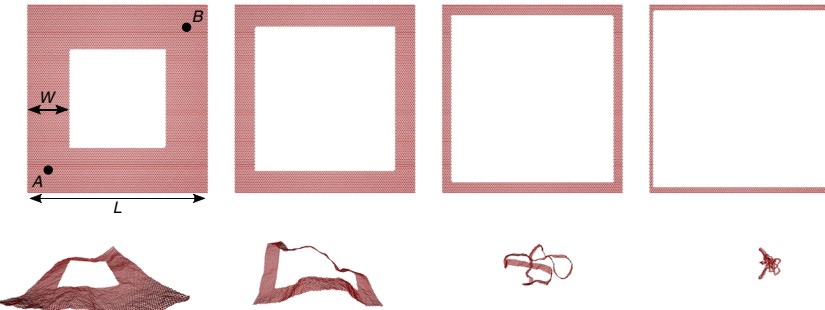

**Fig. 7** Crumpling of a thin frame. The top row shows the initial ($T = 0$) configuration for frames of $L = 100a$ and $W = 24a, 12a, 6a, 3a$ (left to right). The bottom row shows thermalized configurations (for $\tilde{\kappa} = 1.25\,kT$ and $\epsilon = 1800\,kT/a^2$) for each of these geometries, showing a clear crumpling as the frame width $W$ is reduced. Points $A$ and $B$ of the leftmost frame are used to define an order parameter for crumpling in the text

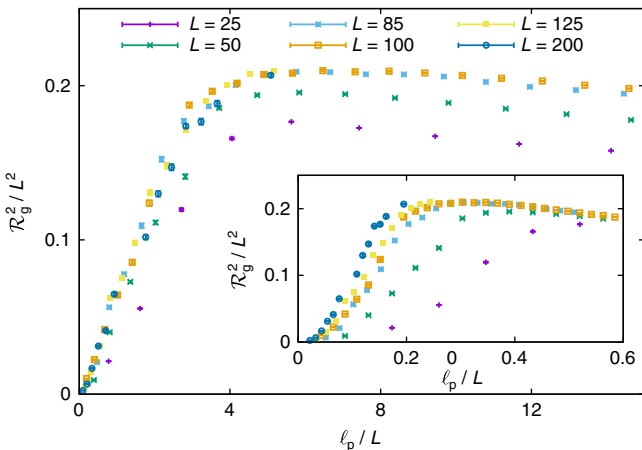

**Fig. 8** Scaling in thin-frame crumpling. We plot the radius of gyration for frames of different $L$ and $W$ against $\ell_p/L$, where the persistence length $\ell_p = 2W\kappa^R(W)/kT$ and the renormalized bending rigidity $\kappa^R(W)$ are defined in Eq. (11). The curves for different system sizes collapse when plotted against this scaling variable. The inset shows that neglecting thermal renormalization of the bending rigidity, that is, considering $\ell_p^0 = 2W\kappa_0/kT$, leads to a poorer collapse. In all these simulations, we have used $\tilde{\kappa} = 1.25\,kT$ and $\epsilon = 1800\,kT/a^2$. All error bars (not visible at this scale in most cases) represent the standard error of the mean

where $\nu$ is the critical exponent describing a normal–normal correlation length that diverges at the crumpling transition. When considering this equation, it is important to notice that, while the exponent is universal, the algebraic prefactor is not and depends on all the parameters. In particular, for a given finite size, the transitions in Fig. 3 seem to be of varying sharpness. However, the values of the critical exponents for the sharpest looking transition (the full membrane) are known from previous work. In the following, we will perform the FSS analysis and a fit to Eq. (8) only for our most perforated membrane (the rightmost curve in Fig. 3). If it is critical exponents turn out to be compatible with those of the full sheet, we can conclude that the intermediate curves will be in the same universality class too.

Figure 6 shows the results of this analysis. We obtain $\theta = 4/d_H + 1/\nu = 2.88(7)$, to be compared with $\theta = 2.86(1)$ from a recent dedicated FSS study for the full membrane[31]. Extracting the values of $d_H$ and $\nu$ separately is more difficult. In principle, one could compute $\nu$ by studying the drift in the position of the peak $T_c^{(L)} \simeq T_c^\infty + AL^{-1/\nu}$, but this has very strong corrections to the leading scaling[31]. Alternatively, one could consider the critical scaling of the specific heat (yielding $\alpha/\nu$ and hence $\nu$ from

hyperscaling), but in this case, one has to include an analytical contribution that introduces an extra fitting parameter: $C_V = C_a + AL^{\alpha/\nu}$. Since, unlike for the full membrane[31], we have to discard sizes $L < 50$ due to finite-size effects, we do not have enough degrees of freedom to obtain a reliable computation of $\nu$. We have checked, however, that the value $\nu = 0.74$ for the standard crumpling transition is consistent with our data (Supplementary Note 2). Using this estimate of $\nu$, we obtain $d_H = 2.62(7)$. In short, the transition in these perforated membranes is compatible with the universality class of the crumpling transition for pristine sheets, even though its location is shifted downward in temperature by an order of magnitude.

**Crumpling of thin frames.** It is illuminating to consider what happens when all perforations are combined to create a thin frame of width $W$ and overall size $L$, e.g., a membrane interrupted by a single large square hole. As shown in Fig. 7 (simulations at fixed temperature and $L$ with varying $W$), there is now a striking crumpling transition as a function of hole size. As an order parameter for this crumpling transition, imagine erecting the normal to these frames at the points $A = (W/2, W/2)$ and $B = (L - W/2, L - W/2)$, where we use an $xy$-coordinate system superimposed on the frame at $T = 0$ with origin at the lower left corner. Then, in the flat phase of the frame (left side of Fig. 7, when the hole is small), we expect in the limit of large frame sizes, $\langle \hat{\mathbf{n}}_A \cdot \hat{\mathbf{n}}_B \rangle \neq 0$. Indeed, in the limit of a vanishingly small hole ($W \rightarrow L/2$), we expect[21]

$$\langle \hat{\mathbf{n}}_A \cdot \hat{\mathbf{n}}_B \rangle_L = 1 - \frac{kT}{2\pi\kappa_0}\left[\eta^{-1} + \ln\left(\frac{\ell_{th}}{a}\right) + C\frac{kT}{\kappa_0}\left(\frac{\ell_{th}}{L}\right)^\eta\right], \quad (9)$$

where $C$ is a positive constant of order unity, $\eta \approx 0.8$ and the thermal length scale is

$$\ell_{th} = \sqrt{\frac{16\pi^3\kappa_0^2}{3kTY_0}}. \quad (10)$$

Thus, $\lim_{L\rightarrow\infty} \langle \hat{\mathbf{n}}_A \cdot \hat{\mathbf{n}}_B \rangle_L \neq 0$, indicating that the normals on diagonally opposite corners are correlated. In contrast, when the frame is crumpled (right side of Fig. 7, when the hole is large), we clearly have $\lim_{L\rightarrow\infty} \langle \hat{\mathbf{n}}_A \cdot \hat{\mathbf{n}}_B \rangle = 0$. In the case of square frames, we can estimate where the transition occurs by comparing the frame size $L$ to the persistence length for thin frames of width $W$[21].

$$\ell_p = \frac{2W\kappa^R(W)}{kT}, \kappa^R(W) = \kappa_0\left(\frac{W}{\ell_{th}}\right)^\eta. \quad (11)$$

Here $\kappa^R(W)$ is the thermally renormalized bending rigidity. Crumpling out of the flat phase should occur when $L > \ell_p$, which

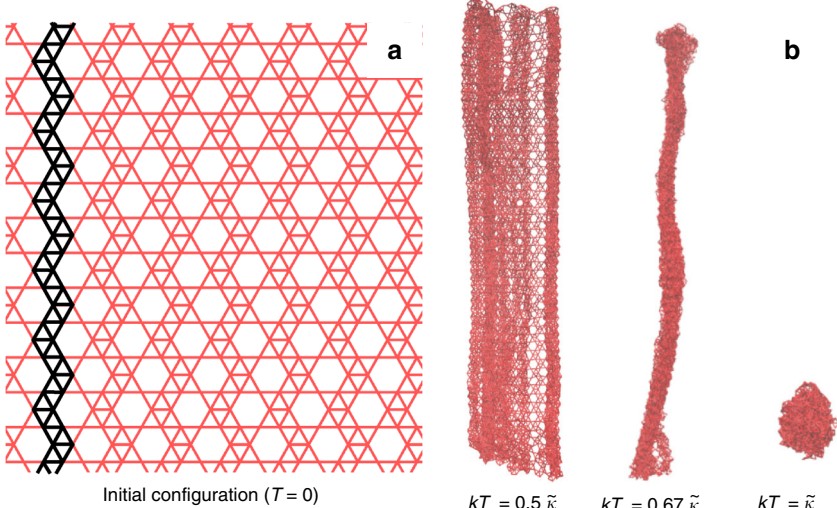

**Fig. 9** Two-step crumpling transition in an anisotropic sheet. The zig-zag pattern of approximately vertical struts reinforced by edge-sharing triangles make this structure more rigid in the vertical than in the horizontal direction (see highlighted example in the figure). We show snapshots of thermalized configurations for several temperatures. As $T$ increases, the anisotropy in the pattern of perforations makes the membrane first fold into a tight cylinder, before crumpling completely. This geometry corresponds to the system labeled Pattern 3 in Fig. 3 and in Supplementary Fig. 1. The $T = 0$ snapshot **a** is a close-up to a $30a \times 30a$ section of the lattice, while the finite-temperature snapshots **b** show the full $100a \times 100a$ system

suggests a scaling form for the radius of gyration of Eq. (4), namely,

$$\mathcal{R}_g^2 = L^2 F(\ell_p/L),\qquad(12)$$

This scaling ansatz (where crumpling is indicated by the behavior for small $x$, $lim_{x\to 0} F(x) \sim x$) is checked for a wide variety of frame dimensions $L$ as a function of $W$ in Fig. 8, which shows excellent data collapse as $L$ becomes large. Note that the collapse is not nearly so good if one simply scales with a bare persistence length (inset), indicating that thermal fluctuations play an important role in our simulations. For this problem, it is known that the crumpled phase is robust to distant self-avoidance[43]. Indeed, the crumpled phase only swells slightly with a scaling function in Eq. (12) that behaves accordingly to $F(x) \sim x^{4/5}$ for small $\ell_p/L$. Of course, considerably more work would be required to demonstrate convincingly that there is a sharp phase transition in the thermodynamic limit. Here, the nontrivial width-dependent scaling of the thermally renormalized persistence length in Eq. (11) suggests that the appropriate limit is $L$, $W \to \infty$, with fixed $W(W/\ell_{th})^\eta/L$. In short, this analysis suggests that there could be a novel transition for single frames, where both a crumpled and flat phase would survive in a polymer-like large-size limit. Even if this transition were simply a crossover, we expect a dramatic change in mechanical properties, such as the response to bending, pulling and twisting, when the frame crumples[21].

We note finally that the crumpling temperature for unperforated membranes can be estimated (up to logarithmic corrections) from Eq. (9) as $kT_c \approx 2\pi\eta\kappa_0$, in approximate agreement with the transition temperature associated with the black curve in Fig. 3.

## Discussion
We have studied the mechanics of thermalized membranes with a dense array of holes and found that the perforations can bring the crumpling temperature into an experimentally accessible regime. From Fig. 5, we have $kT_c/\tilde{\kappa} \simeq A(1-s)^{1.93}$ for the crumpling temperature as a function of the area fraction removed $s$, independent of the detailed arrangement and size of our periodic lattice of holes. In addition, we have found that, with an anisotropic pattern of perforation one can induce a first partial crumpling at an even lower temperature. Indeed, see Fig. 9, a system where the perforations are asymmetric or arranged in such a way that one of the membrane's axes presents less bending resistance will first fold and roll into a very tight cylinder, before crumpling completely. See reviews by Radzihovsky and by Bowick in ref. [23] for a discussion of two-stage crumpling. These observations provide a potential method for bridging the gap between the theoretical expectations for the crumpling transition and the experimentally accessible temperatures.

A subtle issue is our neglect of distant self-avoidance. The nearest-neighbor springs in Eq. (1) embody an energy penalty of order $\epsilon a^2$ when nearest-neighbor nodes overlap, a number that greatly exceeds $kT$. Adding a hard sphere excluded-volume interaction between second-nearest neighbors would create an entropic contribution to $\tilde{\kappa}$ of order $kT$, which might produce a small shift in the crumpling temperature. The existence of a sharp crumpling transition in unperforated membranes with distant self-avoidance remains unclear at the present time[10,44]. The presence of a lattice of large holes will certainly reduce the effect of distant self-avoiding interactions, especially when the removed area fraction becomes large. When distant excluded-volume interactions are non-zero but weak, theory predicts a sharp transition between a low-temperature flat phase and a high-temperature crumpled phase with a nontrivial fractal dimension $d_H \approx 2.5$[45–47], qualitatively similar to the findings for perforated membranes presented here. In addition, we have argued for the existence of a sharp crumpling transition when all perforations are combined to create a thin frame with a single large hole in the center of the membrane. In this case, it is well known that the crumpled ring polymer phase survives the imposition of distant self-avoidance[43]. We hope our results will stimulate allocation of resources (both experimental and computational) that will allow investigations of distant self-avoidance in the presence of a lattice of perforations. Even if distant self-avoidance smears out a sharp crumpling transition, we nevertheless expect qualitatively different mechanical behavior in the regimes identified here for thermalized kirigami sheets.

## Methods

**Our simulations**. We have simulated model (1) for sizes ranging from $L = 25a$ to $L = 150a$ with molecular dynamics in an NVT ensemble, using a standard Nosé-Hoover thermostat[48,49]. All simulations were carried out with the help of the HOOMD-blue package[50,51]. Smaller sizes (up to $L = 50a$) were simulated on CPUs using a message-passing interface (MPI) parallelization, while for larger systems, we have used GPUs. We use a simulation timestep of $\Delta t = 0.0025$ (in natural units where $a = m = kT = 1$). We start with a flat sheet in the $xy$-plane, and add a small random $z$ component to all the nodes, in order to get the molecular dynamics started. We then follow the evolution for $2 \times 10^8$ timesteps, discarding the first 10% for thermalization and using a jackknife procedure[42] to estimate statistical errors. Converted into wall-clock time, $10^8$ steps of a simulation of size $L = 100a$ (with 11,484 nodes, 34,023 bonds and 33,597 dihedral angles) require about 8 h of execution time on an NVIDIA Tesla K40. Our total simulation time has been the equivalent of $\approx 5$ months of a single Tesla K40.

**Data availability**. The data that support the findings of this study are available from the corresponding author upon reasonable request.

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

## Acknowledgements

Work by M.J.B was supported through the NSF DMREF program, via grant DMREF-1435794 and by the KITP grant PHY-1125915. Work by D.R.N. was primarily supported through the NSF DMREF program, via grant DMREF-1435999, as well as in part through the Harvard Materials Science Research and Engineering Center, via NSF grant DMR-1420570. S.B. was partially supported by IGERT DGE-1068780. D.Y. was supported by the Syracuse University Soft Matter program. All authors thank the KITP for hospitality during the completion of some of this work. We benefited from frequent discussions with the experimental groups of P. McEuen and I. Cohen at Cornell University. We also thank Suraj Shankar for valuable discussions. D.Y. acknowledges funding by through contract No. FIS2015-65078-C2-1-P, jointly funded by MINECO (Spain) and FEDER (EU), and the resources and assistance provided by BIFI-ZCAM (Universidad de Zaragoza), where we carried out most of our simulations on the Cierzo supercomputer.

## Author contributions

M.B. and D.N. designed the study; D.Y. performed the simulations and data analysis; S.B. and D.Y. contributed analysis code; D.Y. wrote the paper with contributions from all authors.

## Additional information

**Competing interests:** The authors declare no competing financial interests.

