## [Peer Review File · Nature Communications]

Reviewers' comments:

Reviewer #2 (Remarks to the Author):

In my original report, I judged that "Overall, the manuscript addresses a topic of current interest; the project is well executed and the paper is well written. I recommend its publication in Nature Physics, after the authors have considered my comments."
The authors have indeed considered and responded to these comments. I thus recommend publication of the current manuscript in Nature Communications.

Reviewer #4 (Remarks to the Author):

I have reviewed the revised manuscript and point-by-point response to the original reviewer reports submitted to Nature Communications. The manuscript describes a significant finding on a long standing problem deriving from the fundamental interplay of fluctuations, geometry and elasticity of a basic class of materials, 2D solid sheets. The work is very likely to impact further research into thermal fluctuations of 2D metamaterials from a theoretical direction, and more directly, experimental directions to manipulate structural and mechanical properties of 2D materials, like graphene. Thermal fluctuations are well known to collapse 1D polymeric objects, but to date, the possibility of entropy driven collapse of 2D solids remains only a theoretical possibility. This manuscript provides a very simple and potentially critically enabling direction for experiments to realize this phenomena for the first time, an important result. The manuscript is very clearly written for broad audience in mind. In short, it is well suited in scope and impact for Nature Communications.

I have also reviewed the responses of the authors to the reviewers technical concerns. To my reading, these have been clearly and suitably addressed in the authors revision, with the exception of the final point raised by reviewer 3, who requested that the authors provide some information about the effective elastic constants of perforated sheets. The authors raised the basic concern that the non-trivial effects of thermal fluctuations will ultimately control the transition, and their calculation is beyond the scope of the manuscript. While it is not reasonable to expect a serious analysis of finite temperature elasticity of the sheets as the authors point out, the difficulty to numerically evaluate "bare" elastic constants from simulated $T=0$ deformations of the model systems is rather low, as what originally noted by reviewer 3. Including $T=0$ bending and Youngs modulus, perhaps as inset to Fig 4, would provide at least some evidence that the reduction in crumpling temperature cannot be explained, for example, purely through the drop in the $T=0$ bending modulus, which one expects to be linearly decreasing with removed area

fraction. It is understood that some of the hole patterns break the initial hexagonal symmetry of the mesh and the effective elastic behavior will be more anisotropic, but some reasonable attempt to quantify effective (mean) $T=0$ elastic properties would be valuable and seems appropriate here.

Dear editors,

Please find attached a revised version of our manuscript “Thermal crumpling in perforated two-dimensional sheets”, which we are submitting for publication in Nature Communications. Our work has been reviewed by Referees 2 and 4, whom we thank for their careful reading of our manuscript and for their suggestions. Both referees conclude that the current version of the manuscript is well written and presents well-executed research of broad interest and therefore recommend publication in Nature Communications.

Reviewer #4, however, raises a final point for clarification, which we address below

Reviewer #4: *I have also reviewed the responses of the authors to the reviewers technical concerns. To my reading, these have been clearly and suitably addressed in the authors revision, with the exception of the final point raised by reviewer 3, who requested that the authors provide some information about the effective elastic constants of perforated sheets. The authors raised the basic concern that the non-trivial effects of thermal fluctuations will ultimately control the transition, and their calculation is beyond the scope of the manuscript. While it is not reasonable to expect a serious analysis of finite temperature elasticity of the sheets as the authors point out, the difficulty to numerically evaluate "bare" elastic constants from simulated $T=0$ deformations of the model systems is rather low, as what originally noted by reviewer 3. Including $T=0$ bending and Youngs modulus, perhaps as inset to Fig 4, would provide at least some evidence that the reduction in crumpling temperature cannot be explained, for example, purely through the drop in the $T=0$ bending modulus, which one expects to be linearly decreasing with removed area fraction. It is understood that some of the hole patterns break the initial hexagonal symmetry of the mesh and the effective elastic behavior will be more anisotropic, but some reasonable attempt to quantify effective (mean) $T=0$ elastic properties would be valuable and seems appropriate here.*

Reply: We agree that this is a legitimate point that required some clarification in the text. It can, however, be easily addressed. As explained in a new short Section V at the end of the SI, the zero-temperature bending rigidity is indeed linear in the fraction of removed area. If the onset of crumpling were simply determined by the $T=0$ effective bending rigidity, one would then expect T_c to be also a linear function of $(1-s)$. We obtain instead $T_c \sim (1-s)^{1.93}$: a result which clearly indicates that thermal fluctuations are essential in understanding our result.

The issue of the Young's modulus is more subtle. Indeed, the $T = 0$ elastic response of our perforated sheets to *uniaxial* stresses is going to be both anisotropic and heavily dependent on the details of the perforation pattern. This is not a problem, though, as can be seen in a number of ways:

- From a theoretical point of view, the Young's modulus affects the crumpling transition temperature only as a logarithmic correction. This can be most easily seen by examining eq. (S2) in the SI, which gives the normal-normal correlation function.
- The above-mentioned fact has been verified in the literature. As an example, consider the critical study of Ref. [32], where the authors study full sheets and keep a fixed $\varepsilon = 1$. Their resulting value of the critical temperature is $\kappa / kT_c \approx 0.773$. In contrast, we use $\varepsilon a^2 / \kappa = 1440$. Our resulting value of the crumpling temperature for the full sheet is $\kappa / kT_c \approx 0.19$. A change in ε of more than three orders of magnitude resulted in a change in T_c of a factor of four.
- Finally, we can point to one specific example in our work. As shown in the SI, Patterns 1 and B have almost the same amount of removed area (56% and 53%) and correspondingly very close crumpling temperatures (see Figure 3). However, their $T = 0$ stretching constants are clearly very different.

In short, $\kappa_{\text{eff}}(T=0)$ is essentially the same quantity as the amount of remaining area and cannot explain the non-linear behaviour of $T_c(s)$, while $Y_{\text{eff}}(T=0)$ has very little effect on the crumpling temperature.

In order to explain these issues, we have added a new section to the SI (Section V). We have also added a discussion paragraph after Eq. (7) and plotted the linear dependence of $\kappa_{\text{eff}}/\kappa_0(s)$, to make the contrast with $T_c(s)$ more graphic. All these changes have been highlighted with blue text in the revised version of our manuscript, for easier checking.

We note finally that, when preparing the transfer of our manuscript from *Nature Physics* to *Nature Communications*, we inadvertently submitted a slightly older version of our SI text. The only significant difference was that the originally submitted version did not include Eq. (S2) explicitly (just cited the paper where it was obtained). Since we now make explicit use of (S2) to justify the logarithmic effect of Y on the crumpling temperature, we have taken this opportunity to restore the correct version.

We hope that, with these changes, our manuscript will be found suitable for publication in *Nature Communications*. With our best regards,

D. Yllanes
S. Bhabesh
D.R. Nelson
M.J. Bowick

REVIEWERS' COMMENTS:

Reviewer #4 (Remarks to the Author):

The authors have satisfactorily address the concerns raised in my review. I can recommend the manuscript, in its current form, for publication in Nature Comm.